# The Assessment of Early Server Childhood Caries Status in Abandoned Institutionalized Children

**DOI:** 10.3390/ijerph19148632

**Published:** 2022-07-15

**Authors:** Oana Elena Stoica, Daniela Esian, Anamaria Bud, Alexandra Mihaela Stoica, Liana Beresescu, Cristina Ioana Bica

**Affiliations:** 1Department of Pedodontics, George Emil Palade University of Medicine, Pharmacy, Science, and Technology of Târgu Mureș, 540139 Târgu Mureș, Romania; oana.stoica@umfst.ro (O.E.S.); daniela.esian@umfst.ro (D.E.); anamaria.bud@umfst.ro (A.B.); cristina.bica@umfst.ro (C.I.B.); 2Department of Odontology and Oral Pathology, George Emil Palade University of Medicine, Pharmacy, Science, and Technology of Târgu Mureș, 540139 Târgu Mureș, Romania; 3Department of Preventive and Community Dentistry, George Emil Palade University of Medicine, Pharmacy, Science, and Technology of Târgu Mureș, 540139 Târgu Mureș, Romania

**Keywords:** severe childhood caries, institutionalized children, dmft index

## Abstract

Oral health is a critical indicator of children’s quality of life, which at this early age, depends exclusively on the attention, involvement, and guidance of parents or caregivers. Assessing carious damage and measuring the prevalence of early severe tooth decay in temporary teeth in children is obtained by calculating the dmft index, giving the sum of an individual’s decayed, missing, and filled teeth. The aim of our study was to conduct a clinical examination of the oral status of institutionalized children from Romania. We selected and included in the study 144 children, both boys and girls in equal number, with ages of five or six years old, from which 110 were eligible for the study and met the inclusion criteria. Of all children, 20.90% were five years old girls, 27.27% five years old boys, 26.36% six years old girls, and 25.46% six years old boys. Of all, 10,45% had more than three incisors presenting decays, equally affected by gender. Of the boys’ group, 20% of age six had more de two canines affected, and 26.7% of five years old girls had more than three affected canines. Of the five years boys’ group, 24.3% had more than three affected temporary molars, 44.9% of six years boys had more than five. Of the five years old girls, 33.3% had more than four temporary molars affected and, 56.6% of the six years old girls’ group had more than five molars presenting carious lesions. The dental status and dmft values were similar for five years old and six years old children and similar boys and girls. Due to the vulnerability of young children that consists not only in their inability to identify, express, and address their own needs but also the lack of parental support, lack of an optimal diet for age, and proper hygiene, they reach adolescence with an impaired dental status, inappropriate for their age.

## 1. Introduction

According to the Bangkok IAPD (International Association of Pediatric dentistry), severe early childhood caries (SECC) is defined as the presence of one or multiple carious lesions or fillings on the crown surfaces of any primary tooth in children under six years of age [1,2]. Dental caries is a progressive and irreversible disease and represents the most common cause of morpho-functional imbalance of the stomatognathic system. It affects both permanent and primary teeth, causing an aesthetic-physiognomic discomfort.

The presence of this pathology, especially in young children, reduces the functional and masticatory roles of the dental arches and becomes a chronic source of infection, endangering all vital systems [3].

Primary teeth have an essential role in maintaining space on the dental arches for the future permanent teeth and for the harmonious orofacial development of the child. Even though early severe childhood caries can be prevented, it currently affects more than 600 million children globally and remains an ignored pathology by most parents or caregivers due to the lack of information regarding dental health. Untreated, severe childhood caries have a major impact on the children’s life quality, and also on the whole family, being considered an unnecessary burden on society [4,5,6]. A child with advanced tooth decay cannot eat properly, has poor sleep, slowed growth, sometimes irritability, and when caries becomes complicated, they present decreased ability to concentrate.

Complicated dental caries makes the first visit to the dentist to be one that will leave the child with not very pleasant memories. A child with pain will certainly be uncooperative and may develop anxiety about the doctor, fear that may accompany him throughout life. A tooth affected by caries will end up being extracted. The treatment of a complicated caries requires greater financial and time resources than in the case of the treatment of a simple, incipient caries. Furthermore, the infection developed in dental caries is considered, according to the literature, the outbreak disease of microbial etiology with the possibility of dissemination.

The initiation of the caries affection begins when the intensity of the cariogenic situation in the oral cavity exceeds the resistance of the dental hard tissues. The interaction and intensity of a complex of cariogenic factors leads to the formation of dental caries, and includes tooth susceptibility, health and integrity of dental tissues, bacterial plaque, and fermentable food substrate [7,8,9].

Another key element is the time factor, which is considered very important and can be reduced or interrupted by dental hygiene or improvement in the quality or consistency of food [10]. To trigger tooth caries process, all these factors must interact. The main component is still represented by the bacterial infection, the hydrocarbons being the main cariogenic substrate, thus a diet poor in hydrocarbons qualitatively delays the onset of dental caries [11,12]. Although temporary tooth decay evolves similarly to permanent teeth, clinically, there are a series of particularities determined by the following factors: Structural features at different stages of development of temporary dental tissues, risk factors that favor the appearance of dental caries, the level of damage of the hard tissues and dental pulp reaction [13].

The treatment of the temporary teeth affected by carious lesions is mandatory and it has three main objectives: Stopping the evolution of the caries process, preventing local and remote complications, and preserving dental vitality in order to ensure the physiological rhythm of root resorption to keep the tooth on the arch until the age of exfoliation.

For the incipient active caries (“white spot”), the treatment consists in correcting eating habits, correcting or establishing correct oral hygiene, adapting the methods and techniques in relation to the child’s age, stimulating local remineralization mechanisms through local applications of fluor-based materials. In simple caries with a lack of substance, the following steps are adopted as a therapeutic attitude: Preparation of a cavity, treatment of dentinal wound, and the restoration of coronary destruction using glass-ionomer-based cements. Due to the young age of the little patients, there are a series of peculiarities regarding the structure of the temporary teeth and implications in cavity preparation. When the pulp chamber is more voluminous (age 5–6), with protruding pulpal horns, we have the risk of opening the pulp. Due to the orientation of the enamel prisms, beveling is not necessary after the age of 3–4 years, there may be difficulties in achieving the preventive extension to the class II cavity. The presence of the outer layer of aprismatic enamel makes the demineralization time to be longer when composite materials are used instead of cements.

Very young children who are barely gaining dental experience (for children who are in first contact with the office, there should be no pain at all). For anxious patients with increased fear of dental work and rotary instruments, using manual instruments is mandatory. Complicated caries of temporary teeth is common and is explained by the early appearance of the caries process, the rapid evolution in depth, due to the structural particularities of the temporary teeth and neglecting the treatment of simple caries. To the listed factors can be added chemical, thermal, mechanical, and bacterial irritations produced during dental treatments, such as irrational use of the turbine, failure to observe accidental micro-openings, or trauma [14,15]. Endodontic management is always considered within the overall context of occlusal development, with due consideration to occlusal guidance and space maintenance. Our treatment methods include pulpotomies, pulpectomy procedures, and extractions. As therapeutic agents, formocresol and calcium hydroxide are the materials that are used for pulpotomies, glass ionomers are used as filling material, and we perform metal crowns for the final restoration. For the root canal filling in pulpectomies, we use the Calcipast paste, a resorbable calcium hydroxide iodoform paste.

In the educational system of orphanages, the prevention of oral pathology is not a priority, and there is no precise interest in dental hygiene. Furthermore, because of a deficient financial system, the food that institutionalized children receive and consume is not qualitative, based on carbohydrates and sugar. The diet in orphanages or abandoned children’s homes is based more on quantity due to the large number of children who live in these places, therefore, the nutritional value and fiber content are extremally reduced [16]. Abandoned children who are not supported by their parents or caregivers and who do not have knowledge regarding dental hygiene will not pay attention on their own initiative to teeth brushing, so their dental status is precarious. They have no interest in improving hygiene, not knowing the possible risks, and in most cases, their first visit to the dentist is often around the age of adolescence [17,18].

Data from the National Authority for the Protection of the Rights of the Children and Adoption show that in the Romanian protection system, there are currently 52,783 children, of which 17,096 in the residential system (orphanages) and 35,687 in the” family” system, which includes 17,835 foster cares, and 13,133 extended families. Out of the total reported, approximately 76% are children abandoned at birth, and 89% spend their first 18 years, and 95% spend their six years of life in these places [19].

Therefore, the purpose of our study was to assess the dental status of abandoned and institutionalized children up to six years of age in order to improve the educational system and support dental prevention in these disadvantaged cases [20,21,22].

## 2. Materials and Methods

### 2.1. Study Design

Our study was carried out over a period of 2 years and 6 months, starting from September 2018 until March 2021, in the Private Dental Clinic StoicaMed from Targu-Mures, Mures County. The methodology and design of the study were approved by the ethics committee of the clinic (no. 37/15 September 2018). In order to begin the study, we obtained the written approval consent of the legal guardians of the institutionalized children, but also of the managers of these centers. The clinical examination and data recording was performed by 4 specialized dentists, with more than 4 years of experience in pedodontics. We selected and included in the study144 children, both boys and girls in equal number, meeting the following criteria of inclusion: Ages of 5 or 6 years old, the presence of temporary teeth in complete formula, clinically healthy, their presence in the orphanage for at least 4 years. The exclusion criteria were: Children under 5 years of age or older than 6 years, children with general or systemic problems, children who have recently been institutionalized (Figure 1). Regarding the section of general and systemic problems, the list included: Associated psychiatric and mental disorders, genetic syndromes, multisystemic pediatric inflammatory syndrome, and congenital disorders.

### 2.2. Clinical Examination and Assessment

Clinical dental examinations were performed using disposable dental instruments (mirrors and dental probes). For each young patient, the dmft (decay, missing, filled, tooth) index was assessed and calculated. When written in lower case, the dmf index is a variation of DMFT index (used for permanent dentition) that applies to the primary dentition and it represents the total number of teeth that are degraded/decayed, missing, or filled, with scores between 0 and maximum 20 for each child. Due to the difficulty of distinguishing between teeth extracted due to caries and those that have exfoliated naturally, missing teeth can be ignored according to our protocols.

### 2.3. Statistical Data Analysis

Jamovi statistic software (2.2.5.0) (Sydney, Australia, https://www.jamovi.org/download.html, accessed on 7 June 2022) for Windows 2021was used for the data descriptive analysis, including independent *t*-tests. For the significance level, the value of 0.05 was chosen, and *p* was considered significant if *p* ≤ 0.05.

## 3. Results

Of 144 children initially included in the study, 23 were excluded due to not meeting our inclusion criteria and 11 children did not cooperate and could not be examined. The following Table 1 presents the division of the total 110 eligible children into groups according to gender and age.

Therefore, the study included 20.90%—5 years old girls, 27.27%—5 years old boys, 26.36%—6 years old girls, and 25.46%—6 years old boys.

Of all children, a total of 64 (58.2%) had the frontal incisors intact (*p* < 0.01), 11 (10%) presented decays on 2 incisors (*p* < 0.01), 23 (20.9%) had decays on 4 incisors, 9 (8.2%) had 6 incisors affected, and 3 (2.7%) presented carious lesions on 8 incisors (*p* < 0.01).

All the results of the and the frequency of the decidual teeth affected are presented in the following Table 2, Table 3 and Table 4.

For the number of temporary incisors affected, we calculated a mean difference value of 1.745, for the temporary canines, a mean difference value of 0.818, and for the temporary molar group, the mean difference value was 3.827.

Regarding the results for the canine examination, 68 children (61.8%) had all canines intact, 39 children (35.5%) presented caries on 2 different canines, and 2.7%, meaning 3 children presented carious lesions on all 4 canines, as shown in Table 3.

In means of the caries incidence for the decidual molars, after examination, we counted 7 children (6.4%) with intact molars, 14 children (12.7%) with 2 affected molars, 31 children (28.2%) presenting 3 molars affected by decay, 32 (29.1%) with 4 molars with carious lesions, 10 children (9.1%) presenting 5 affected molars, 3 children (2.7%) with 6 decidual molars affected by decay and 13 children with all 8 molars affected. The results and the frequencies are presented in Table 4.

Dmft was calculated for each child after the dental examination, with the lowest score of 0 calculated for 7 children, the highest score of 20 calculated for 3 children. We have calculated a score of 3 for the dmft index in 11.2% cases, score of 4 in 11.2% also, score of 5 in 9.3% cases, score of 6 in 12.1% cases, score 7 in 8.4%, score 8 in 14%, score 9 in 9.3%, score 12 in 5.6% and 14 in 2.8% cases. The results and the frequencies are presented in Table 5.

We present one of the most severe clinical cases, 5 years and 7-month year old boy, generally and clinically healthy, presenting, after oral examination, a dmft score of 20 (Figure 2). The carries lesions occur symmetrically along the arches, affecting all decidual teeth. Due to the lack of treatment and the aggressive evolution of the decay, the crowns for all upper and lower incisors and canines are destroyed. The decidual molars present on mesial, distal, vestibular, and occlusal surfaces, with multiple caries affecting the undermining marginal enamel ridges.

## 4. Discussion

All children included in the study were five to six years old, an age extremely sensitive regarding oral health and oral habits, oro-maxilar development, mastication habits, and type of food and drinks preferred and consumed. During this time interval, caries receptivity is at its very high and the parents’ role in maintaining and protecting their children’s health is essential [23,24].

In Romania, the institutions that shelter abandoned children are not a national priority and the national financial funds often cover only the basic living of the children. From the beginning, the situation is deficient due to the low ratio of employees to the orphanage and the high number of children of various ages. Epidemiological studies have shown that the majority of children (75%) are abandoned at birth and 25% of institutionalized children are abandoned in the first three years of life. In recent years, the number of abandoned children has increased, being cared for by the same number of employees or volunteers of orphanages. On average, it was reported that a caregiver is responsible for eight or more institutionalized children.

The food consumed by these children is often quantitative in the deficit of the nutritional value necessary for a child in full development and rich in sugars and refined carbohydrates. Being the most dangerous type of alimentation, not only for the stomach but also the teeth, such foods cause the formation of bacteria and generally an acidic environment that affects the teeth and gums [25,26]. The effect of a predominantly carbohydrate and refined sugar diet on children with temporary or mixed teeth is well studied by many authors, confirming the essential role of the quality of food consumed by children, both in general development and in the health of dental structures [27].

The incidence of temporary tooth decay in children included in our study is high, similarly affecting both boys and girls, all having the same diet and consuming the same type of food. There is a lack of protocols or education programs in basic or dental hygiene or direct financing for the purpose of purchasing toothbrushes or other accessories. According to other studies conducted in Romania, 40% of children aged between five and seven years, from social backgrounds considered above average, compared to the standard of living in Romania, have tooth decay on temporary teeth. Although the first permanent molar at the age of six is freshly erupted, a study showed damage with tooth decay in 36% of 7-year-olds. Moreover, pre-schoolers with caries are 30% more likely to develop the condition on permanent teeth. In addition, in the world, tooth decay is still the leading cause of tooth loss among children.

Many Romanian parents believe that baby teeth should not be treated because they will be replaced by permanent ones anyway. However, in addition to the importance they have in nutrition, speech, aesthetics, temporary teeth have an essential role in the correct guidance of the eruption of permanent teeth, which are already developing in the maxillary bone.

There are many volunteers in the dental field who try to cover this deficit by visiting orphanages to train children in improving dental hygiene, but their efforts are often insufficient. Our findings are similar to previously published research showing that children who do not benefit from parental guidance and help with their hygiene habits usually suffer a high incidence of carious lesions and further develop mastication deficits and even nervous tics to cover their dental problems [28,29].

Children with severe early childhood caries grow at a slower rate than caries-free children. In addition, the appearance of discoloured, decayed, or absent teeth for any person, adult or child, is associated with an important negative connotation in our culture, being associated with the unhygienic, undesirable, poor lifestyle on the fringes of society. With a maximum dmft of 20, three children from our study show the lack of oral hygiene associated with an incorrect diet and poor dental protocols, examinations, and treatments for these categories of people from our country.

Most studies concluded that by the age of two and a half, the smooth faces of the upper incisors and the occlusal faces of the first molars are affected, and by the age of three and a half, caries progresses on the smooth faces of the upper canines and the occlusal faces of the two temporary molars. At the age of five, the proximal surfaces of all temporary molars are usually affected. If dentists do not intervene in time with the necessary treatments and with prevention, these children reach adulthood with edentations following the extractions, and not having the culture of the importance of oral health [30]. Early severe childhood caries is also known as tooth decay in children, one of the most common dental problems in children [31]. It is an acute, early-onset, rapidly evolving caries that affects the young child. It is manifested by cavities or non-cavities, missing teeth, or blocked surfaces on any temporary tooth as a result of tooth decay [32].

The civilization and information level are constantly growing and contrary to expectations, tooth decay in children has increased greatly lately. A study from 2017 in Romania conducted on children grouped according to the degree and characteristics of the population in our country showed that there are three trends of cariogenic activity with significant differences between them, which become evident at 5–7 years and remain in the same proportion until the age of 13–14 years, namely: The group with intense carious activity, involving approximately 17.3%. The growth rate of caries varies for both sexes around the average of two to three caries areas for each year of age, except for the period 10–11 years. When they reach values of 10 decayed surfaces for girls and 7 for boys.

Numerous studies and authors have found that the introduction of toothbrushing at a younger age and the supervision of brushing by parents significantly reduce the severity of carious damage in children who use incorrect feeding and care habits [33,34,35]. It is recommended that toothbrushing begin as soon as possible after the eruption of the first temporary teeth. Children who have already established the habit of brushing their teeth at the age of one are more likely to be free of caries at the age of three years [36]. Following the discussion with the children who participated in our study and following the way they perform dental brushing, we found that most do not realize the need for dental hygiene, a small part of them brushes their teeth once a day, in the evening, and the rest use a toothbrush once every two to three days, of course without performing a correct brushing as a technique or duration. For young children and pre-schoolers, brushing their teeth does not ensure proper tooth cleaning. By supervising the brushing, the deficiencies of the brushing technique performed by the child are removed. Detection of early severe caries in the early stages of evolution is difficult because white demineralization spots are difficult for parents to notice [37]. The American Academy of Pediatric Dentistry recommends that around the age of one, the child should have the first consultation performed by the pediatric dentist, which in the case of abandoned and institutionalized children is often impossible [38].

Initiated in 1995, the “Overland for Smile” project aims at free dental care for institutionalized children around the world, exchange of information regarding the prevention of dental diseases with people who are in constant contact with children, and the promotion of building experience between dental associations and profile universities between Romania and the neighbouring countries. Over the past six years, mobile caravans have travelled more than 1.5 million kilometres through the most inaccessible areas in 121 countries.

The “Overland for Smile” campaign is carried out under the high patronage of the Italian Embassy in Bucharest, UNIMPRESA Romania, and with the support of the National Union of Dental Associations in Romania. The caravan’s mobile cabinet includes three complete units equipped with five instruments with a self-disinfecting system, laser equipment, ultrasound, air conditioning, and a video camera through which the entire operation can be watched live on the Internet. The treatment space has been designed in such a way that the children feel relaxed through colours, lights, and music, or projections of animated films.

In 2018, a mobile dental caravan was set up, operating in Romania during the summer, focusing on the counties located in the south of Romania, where it reaches over 1800 children from disadvantaged areas. There were also a series of mixed teams made up of volunteer dentists and young students from the Universities of Dentistry, who consult and treat the dental diseases of institutionalized children and initiate dental prophylaxis. The campaign is intended for children in social care centres and focuses on an educational program, informing tutors aiming to improve oral hygiene techniques and the use of oral hygiene aids in order to support the oral hygiene of children.

In addition to this mobile caravan and voluntary charity, the Romanian state offers funds to private offices around the country to treat children up to 18 years old for free. It is necessary that these funds be addressed mainly for these children and to organize group travel mobilities for consultation and treatment in these offices. At the national level, these programs can be improved from an educational point of view, and we consider the idea of emphasizing prophylaxis and maintaining health to be ideal. By continuing our studies in this area and disseminating our results, we sound the alarm on the reality of the situation related to the dental status of institutionalized children, and we will support by all possible means the improvement and progress in this oral medical field [39,40].

## 5. Conclusions

Abandoned and institutionalized children in state orphanages in Romania are a special socially disadvantaged category of the population both in terms of education and intellectual development, basic standard of living, physical development, oral hygiene, and access to medical education. Not benefiting from parental support, having a nutritionally deficient diet rich in refined sugars, the dental status of these children is represented by an increased incidence of severe carious lesions with repercussions on their physical and mental development. Given the results of this study and the dental clinical situation of disadvantaged children, we suggest that there is a need for additional studies in this field, with the possibility of initiating national programs to help and support children’s health.

## Figures and Tables

**Figure 1 ijerph-19-08632-f001:**
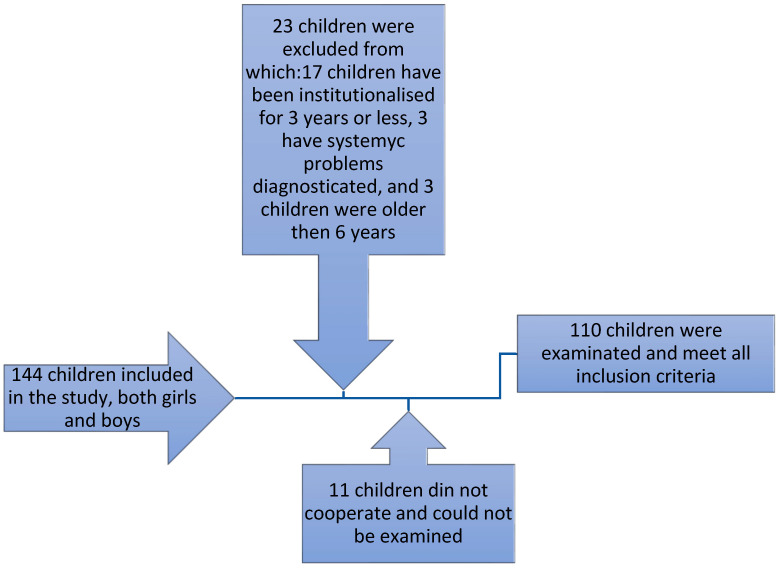
Diagram illustrating the selection of the young patients included in the study.

**Figure 2 ijerph-19-08632-f002:**
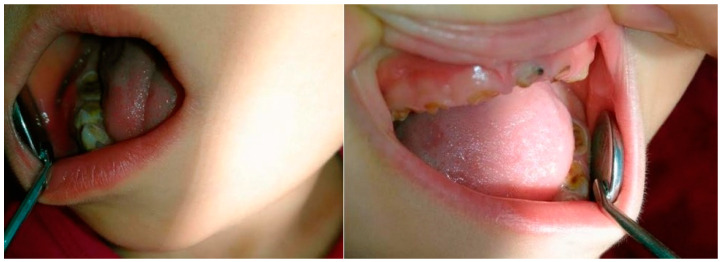
Five years and 7-month year old boy, presenting dmft score 20.

**Table 1 ijerph-19-08632-t001:** One hundred and ten eligible children group division according to gender and age.

Total 110	5 Years Old	6 Years Old
Girls 52 (47.27%)	23 (20.90%)	29 (26.36%)
Boys 58 (52.72%)	30 (27.27%)	28 (25.46%)

(No significant differences between the no. of children included in the groups, divided by age and gender *p* > 0.05).

**Table 2 ijerph-19-08632-t002:** Frequencies of the number of temporary incisors affected for every child.

No. of Incisors Affected/Child	Children	% of Total
0	64	58.2%
2	11	10.0%
4	23	20.9%
6	9	8.2%
8	3	2.7%

**Table 3 ijerph-19-08632-t003:** Frequencies of the number of temporary canines affected for every child.

No. of Canines Affected/Child	Children	% of Total
0	68	61.8%
2	39	35.5%
4	3	2.7%

**Table 4 ijerph-19-08632-t004:** Frequencies of the number of temporary molars affected for every child.

No. of Molars Affected/Child	Children	% of Total
0	7	6.4%
2	14	12.7%
3	31	28.2%
4	32	29.1%
5	10	9.1%
6	3	2.7%
8	13	11.8%

**Table 5 ijerph-19-08632-t005:** Frequencies of dmft index.

Dmft	Children	% of Total	Cumulative %
0	7	6.5%	6.5%
2	7	6.5%	13.1%
3	12	11.2%	24.3%
4	12	11.2%	35.5%
5	10	9.3%	44.9%
6	13	12.1%	57.0%
7	9	8.4%	65.4%
8	15	14.0%	79.4%
9	10	9.3%	88.8%
12	6	5.6%	94.4%
14	3	2.8%	97.2%
20	3	2.8%	100.0%

## Data Availability

All data regarding this manuscript can be checked with corresponding authors.

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
