# Peer review of "The Assessment of Early Server Childhood Caries Status in Abandoned Institutionalized Children"

_ijerph, 2022, doi:10.3390/ijerph19148632_

Round 1

Reviewer 1 Report

1. The research conducted by the group is interesting but needs more insights. The authors discussed that "food rich in sugars and refined carbohydrates" as the reason for poor dental health. Data should be given about the brushing habits of these children. The authors should also address the level of decay for each type of tooth discussed.

2. Minor punctuation errors need to be addressed.

3.  The discussion section should be split into sections.

4. It will be interesting if the authors included in the discussion section what strategies were undertaken to treat the infected teeth of these underprivileged children.

Author Response

Dear Reviewer,

We would like to thank you for taking the time to revise our article, and for the valuable guidelines and improvement possible. We attach our response.

Thank you!

Reviewer 2 Report

Dear Authors, this paper treats about a very interesting subject.

Abandoned institutionalized children need to be treated, if they present any physical or mental concern, and they need to get access to public health as well as children who are not institutionalized. This represent a common problem all over the world and, assessing this problem from the caries point of view is really interesting.

Some issues need to be assessed before the publication of this paper. 

Abstract: Results in abstract are not clear explained. Please explain better your findings in a synthetic way.

Introduction: Line 46-47: please explain better this concept, what is the impact on quality of life due to severe caries?

Line 60-61: it is stated that differences appear clinically from permanent teeth caries and deciduous teeth caries: please explain what are these differences or erase this sentence.

Introduction overall is missing in important part about the suggested treatment of caries disease in children: please add a small chapter about that. References like this could help: Ludovichetti FS, Stellini E, Signoriello AG, DI Fiore A, Gracco A, Mazzoleni S. Zirconia vs. stainless steel pediatric crowns: a literature review. Minerva Dent Oral Sci. 2021 Jun;70(3):112-118.

Materials and methods: Material and methods are overall well written and easily understandable from the reader.

Line 96: it is stated that in the exclusion criteria "children with general or systemic problems" were excluded: please explain better, mental diseases? physical inabilities? systemic health problems?

Results: no issues need to be addressed, they are well written and well understandable

Discussion: For a study like this,  this discussion is really short. What is currently written is actually correct, but it is recommendable to improve your discussion adding some chapter.

Please add a small chapter about the possible treatments available when children are being institutionalized. These references could help:

Duangthip D, Chen KJ, Gao SS, Lo ECM, Chu CH. Managing Early Childhood Caries with Atraumatic Restorative Treatment and Topical Silver and Fluoride Agents. Int J Environ Res Public Health. 2017 Oct 10;14(10):1204.

American Academy of Pediatric Dentistry Council on Clinical Affairs. Policy on Early Childhood Caries (ECC): Unique Challenges and Treatment Options. Pediatr. Dent. 2016, 38, 55–56.

Limitation of the study: please, erase this chapter. What is stated here it is already clear in text.

Conclusions: Please add a small sentence where you state the importance of doing more studies like this not just in romania but also in other countries

Author Response

(The authors gave the same response as above.)

Reviewer 3 Report

This is a weak study. It is just how many kids were with caries in that hospital. Small group and no control group. I doubt whether the authors are able to add control group and describe the results in a more clear manner.

Unfortunately, there is no control group, it is just the oral situation of children of 5 or 6 years who are institutionalised in a hospital in Romania. A control group, either a real control group or a group from the literature, has to be added. Moreover, the statistics have also to be rewritten. Do not start with a p value, but just add such a p value (when applicable) between bracktes at the end of the sentence. Also, the readership does not have much of knowing a mean difference, you should write also what this difference mean and the name the original values. In the tables, also name, with a zero, the missing/not existing number of teeth.

Author Response

(The authors gave the same response as above.)

Reviewer 4 Report

Dear authors, attached are the recommendations considered necessary for your article. Congratulations on your work and good luck!

Author Response

(The authors gave the same response as above.)

Round 2

Reviewer 2 Report

Dear Authors,

the manuscript now is really well written and ready to be published in my opinion.

You did not add any part in the Introduction about the suggested treatment of caries disease in children like it was asked in the previous revision: please add a small chapter about that. References like this could help:

Ludovichetti FS, Stellini E, Signoriello AG, DI Fiore A, Gracco A, Mazzoleni S. Zirconia vs. stainless steel pediatric crowns: a literature review. Minerva Dent Oral Sci. 2021 Jun;70(3):112-118.

Duangthip D, Chen KJ, Gao SS, Lo ECM, Chu CH. Managing Early Childhood Caries with Atraumatic Restorative Treatment and Topical Silver and Fluoride Agents. Int J Environ Res Public Health. 2017 Oct 10;14(10):1204. doi: 10.3390/ijerph14101204. PMID: 28994739; PMCID: PMC5664705.

Author Response

Dear Reviewer,

We would like to thank you again for taking the time to revise our manuscript.

We attach our response.

Have a nice day!

Reviewer 3 Report

While you made considerable efforts to improve the manuscript, still the lack of a control group (even from the literature, only general references are given) and the description of the results remains poor. 

Author Response

(The authors gave the same response as above.)
